# An Evaluation of Omega-3 Status and Intake in Canadian Elite Rugby 7s Players

**DOI:** 10.3390/nu13113777

**Published:** 2021-10-25

**Authors:** Ashley Armstrong, Anthony J. Anzalone, Wendy Pethick, Holly Murray, Dylan T. Dahlquist, Andrew T. Askow, Jeffery L. Heileson, Lyn M. Hillyer, David W. L. Ma, Jonathan M. Oliver

**Affiliations:** 1Canadian Sport Institute Pacific, Victoria, BC V9E 2C5, Canada; wpethick@csipacific.ca (W.P.); hmurray@csipacific.ca (H.M.); 2Wake Forest School of Medicine, Winston-Salem, NC 27101, USA; aanzalon@wakehealth.edu; 3Red Bull Athlete Performance Center, Los Angeles, CA 90404, USA; dylan.dahlquish@redbullperformance.com; 4Department of Kinesiology and Community Health, University of Illinois at Urbana-Champaign, Urbana, IL 61801, USA; askow2@illinois.edu; 5Department of Health, Human Performance and Recreation, Baylor University, Waco, TX 76798, USA; Jeffery_Heileson@baylor.edu; 6Department of Human Health and Nutritional Sciences, University of Guelph, Guelph, ON N1G 2W1, Canada; lhillyer@uoguelph.ca (L.M.H.); davidma@uoguelph.ca (D.W.L.M.); 7The Sport Science Center, Texas Christian University, Fort Worth, TX 76129, USA; jmoliverphd@gmail.com

**Keywords:** eicosapentaenoic acid (EPA), docosahexaenoic acid (DHA), alpha-linolenic acid (ALA), rugby, head impacts, concussion, sub-concussion

## Abstract

Background: EPA and DHA n-3 FA play crucial roles in both neurological and cardiovascular health and high dietary intakes along with supplementation suggest potential neuroprotection and concussion recovery support. Rugby athletes have a high risk of repetitive sub-concussive head impacts which may lead to long-term neurological deficits, but there is a lack of research looking into n-3 FA status in rugby players. We examined the dietary n-3 FA intake through a FFQ and n-3 FA status by measuring the percentage of n-3 FA and O3I in elite Canadian Rugby 7s players to show distribution across O3I risk zones; high risk, <4%; intermediate risk, 4 to 8%; and low risk, >8%. Methods: n-3 FA profile and dietary intake as per FFQ were collected at the beginning of the 2017–2018 Rugby 7s season in male (*n* = 19; 24.84 ± 2.32 years; 95.23 ± 6.93 kg) and female (*n* = 15; 23.45 ± 3.10 years; 71.21 ± 5.79 kg) athletes. Results: O3I averaged 4.54% ± 1.77, with female athlete scores slightly higher, and higher O3I scores in supplemented athletes (4.82% vs. 3.94%, *p* = 0.183), with a greater proportion of non-supplemented athletes in the high-risk category (45.5% vs. 39.1%). Dietary intake in non-supplemented athletes did not meet daily dietary recommendations for ALA or EPA + DHA compared to supplemented athletes. Conclusions: Overall, despite supplementation, O3I score remained in the high-risk category in a proportion of athletes who met recommended n-3 FA dietary intakes, and non-supplemented athletes had a higher proportion of O3I scores in the high-risk category, suggesting that dietary intake alone may not be enough and athletes may require additional dietary and n-3 FA supplementation to reduce neurological and cardiovascular risk.

## 1. Introduction

Rugby is the most popular full-contact team sport in the world, with popularity growing in North America since the debut of Rugby 7s at the 2016 Sumer Olympic Games [1,2,3,4]. Compared to the traditional game of Rugby 15s, Rugby 7s is a shorter game, played over two, 7 min halves, and typically played in a multigame-tournament structure extending over 2–3 days [1]. At the international elite level, tournaments are typically paired, in which two tournaments are played within 10 days [2]. The increase in popularity has led to a direct increase in injuries associated with Rugby 7s, especially concussions, where there are 8.3 concussions/1000 player hours and 7.7 concussions/1000 player hours in Rugby 7s and 15s, respectively [2,3]. Fuller et al. [1] examined injury rates during the 2014/2015 and 2015/2016 Sevens World Series and found that the number one and two most reported injuries throughout a given season were that of concussion-related insults [1]. When compared to other sports, the reported concussion incidence in Rugby 7s has been demonstrated to be lower compared to American-style football, but higher than that seen in ice hockey [2].

Omega 3 fatty acids (n-3 FA) have been reported to play an important role in cardiovascular, cognitive health and neurological development [5], especially eicosapentaenoic acid (EPA) and docosahexaenoic acid (DHA). DHA is particularly rich in the neural and retinal tissue found in the brain [5]. In the last decade, n-3 FA consistently demonstrates that treatment with DHA prior to injury attenuates axonal damage following traumatic brain injuries [6,7,8,9]. Fatty fish such as salmon, mackerel, sardines, herring, and trout are some of the most abundant sources of DHA and EPA [5], and international health authorities have recommended individuals to consume fish 1–2 times a week [10,11]. Unfortunately, dietary consumption is quite low, particularly in North America [12,13] and in athletic populations [14,15], where data from the National Collegiate Athletics Association (NCAA) show US collegiate athletes are well below dietary recommendations (500 mg·day^−1^ DHA + EPA [16]), with n-3 FA intakes averaging 100 mg·day^−1^ [14]. Conversely, the International Olympic Committee (IOC) recommends elite athletes acquire 2 g·day^−1^ of n-3 FA from a food source or supplement with no specifications on EPA or DHA concentration [17]. There is still little consensus on n-3 FA supplement intake, especially in athletic populations.

The Omega-3 Index (O3I) represents the percentage of DHA and EPA making up total erythrocyte fatty acids, with cut-off points of 4% or less, between 4 and 8% and greater than 8% defined as high, moderate, or low coronary heart disease (CHD) risk, respectively [18,19]. In the general Canadian population, mean O3I levels were 4.5%, with less than 3% of Canadians between 20 and 79 years old having levels with low risk for CHD [20]. Specific to athletes, mean values in male and female elite endurance athletes were 4.97 ± 1.19% [21]; and in NCAA male football players, 34% of athletes had levels below 4%, where 66% were between 4% and 8%, and the mean O3I was 4.4% ± 0.8% [22]. Similar results were also seen in a wider spectrum of NCAA athletes, with O3I scores of 4.79% (4.37–5.25%) in males and 4.75% (4.50–5.01%) in females [14]. To date, no authors have examined the O3I or n-3 FA consumption of elite male and female Rugby Union players. It is of particular interest to determine if there are sex differences associated with n-3 FA intake and status.

The purpose of this study was to examine n-3 FA status as measured by the percentage of n-3 FA and O3I, as well as the dietary consumption of n-3 FA through a food frequency questionnaire (FFQ) among elite national Canadian Rugby 7s players. A second objective of this study was to determine if there were any notable sex differences in n-3 FA intake and status.

## 2. Materials and Methods

### 2.1. Subjects

The men’s and women’s National Canadian Rugby 7s teams were contacted to participate in this study. Eligibility criteria included being a current Athlete Assistance Program Carded athlete, a member of the National men’s or women’s Canadian 7s team, over the age of 18, and injury free including not being diagnosed with a concussion. A total of 34 athletes met with criteria and were analyzed: 19 males (24.84 ± 2.32 year) and 15 females (23.45 ± 3.10 year). All athletes who met this criterion had competed at an international Rugby 7s level for more than 1 year. Written and verbal informed consent as documented by Texas Christian University Institutional Review Board (#1706-066-1706) was obtained from all athletes prior to participation.

### 2.2. Study Design

This was a retrospective, cross-sectional, observational investigation conducted in August 2017 during the men’s and women’s Canadian Rugby 7s pre-season prior to the 2017–2018 HSBC World Rugby Sevens Series. Athletes consenting to participate had venipuncture whole-blood samples collected and completed a 21-item n-3 FA FFQ.

### 2.3. Food Frequency Questionnaire

A 21-item FFQ that quantified intakes of alpha-linolenic acid (ALA), EPA and DHA was administered to participants by a Registered Dietitian. The FFQ was developed from the National Cancer Institute’s Diet History Questionnaire [14,15,23]. The FFQ took participants approximately 5 min to complete and was collected at the beginning of their pre-season. The FFQ inquired about intake of foods high in n-3 FA, including fish, seafood, walnuts, flaxseed, flaxseed oil, cod liver oil and canola oil consumption.

For each food, participants also reported the frequency of consumption over the past six months from never to several times a month, week or day. If the food was consumed, participants then recorded the average portion size consumed based on sex-specific portion size from small to large listed in ounces for fish and seafood, cups for walnuts, and teaspoons for oils and flaxseed.

ALA, EPA and DHA intakes (mg/day) were calculated and stratified based on sex, frequency, portion size, and n-3 FA food choice content. Average daily intake of n-3 FAs was used to determine if athletes met the IOC recommendation of 2 g·day^−1^ of n-3 FA from a food source or supplement or the much more defined Academy of Nutrition and Dietetics recommendations of 500 mg·day^−1^ DHA + EPA. The developers of the FFQ [23] provided a database of n-3 FA content of food choices which was derived from previously published databases [23,24,25,26].

The FFQ also inquired about the type and dose of n-3 FA dietary supplement. All athletes were from the same National Sport Organization and those who indicated that they were taking an n-3 FA supplement were all using the same brand (Klean Omega, Klean Athlete, Pittsburgh, PA, USA), recommended by their sport dietitian, as it follows the NSF Certified for Sport program to help ensure products are free of banned substances.

### 2.4. Blood Collection and Preparation

Blood sampling procedures mimicked that of previous literature [27,28]. Blood samples were collected from participants via venipuncture from the antecubital fossa region using standard, sterile phlebotomy procedures, and collected in in spray-coated K2 ethylenediaminetetraacetic acid vacutainer tubes (BD Diagnostics, Franklin Lakes, NJ, USA). Within 30 min of collection, samples were centrifuged at 2000× *g* for 30 min at 4 °C (Fisher Scientific Centrific Centrifuge, Model 225A Cat #04-978-50A, Duuque, IA, USA). Aliquots were collected using a sterile pipette from center of the red blood cells pack within the ethylenediaminetetraacetic acid vacutainer tubes and were transferred to pre-labelled polypropylene vials which were stored at −80 °C until analysis (Environmental Equipment, So-Low, Model C85-5, Cincinnati, OH, USA).

### 2.5. Blood Analysis

Lipids were extracted from plasma by the Folch method, which has been previously described [29,30]. In brief, 50 μL of plasma was mixed with 3 mL of 2:1 chloroform:methanol (v:v) which contained 10 μg of C19:0 (internal standard). Samples were vortexed for 1 min and then 550 uL of 0.1 M KCl was added to each tube with brief vortexing to mix. The tubes were spun at 357× *g* for 10 min at 21 °C to separate phases. The lower phase was extracted and dried under nitrogen gas. Samples were methylated in 300 μL of hexane and 1 mL of 14% boron trifluoride methanol for 1 h at 100 °C. Methylation was terminated by the addition of 1 mL dH20 and 1 mL hexane. The tubes were spun at 357× *g* for 10 min at 21 °C to separate phases. The top layer was extracted and dried under nitrogen gas. Samples were reconstituted in 400 μL of hexane for fatty acid analysis by gas chromatography.

Fatty acid methyl esters were separated and identified on an Agilent 7890-A gas chromatograph equipped with a flame ionization detector and separated on a SP-2560 fused silica capillary column (100 m × 0.25 mm i.d × 0.2 μm film thickness) (Sigma, Cat #24056). The inlet (pressure of 19.5 psi and a hydrogen flow of 10.2 mL/min) and detector were set at 250 °C (hydrogen flow of 30 mL/min, air flow of 450 mL/min and nitrogen flow of 10 mL/min). A volume of 1 μL of sample was injected in the split mode (5:1). The oven was initially set at 60 °C, then it was increased by 13 °C/min to 170 °C and held for 4 min, then by 6.5 °C/min to 175 °C with no hold, then by 2.6 °C/min to 185 °C with no hold, then by 1.3 °C/min to 190 °C with no hold, then by 13.0 °C/min to 240 °C and held for 25 min. The total run time was 49.78 min.

Peaks were identified using OpenLab CDS EZCrome (Agilent, Edition A.04.06 version 1.255.227, 2014) by comparing peaks to authentic FA methyl ester standards (NuChek, Elysian, MN, USA). C19:0 internal standard was used to calculate absolute FA concentrations (μg/mL) and relative percent composition (%) of individual FA was determined from the relative integrated area under the curve for all identified FA.

### 2.6. Statistical Analysis

Data were analyzed using IBM Statistical Package for the Social Sciences IBM SPSS Statistics software (version 27, Armonk, NY, USA). Descriptive statistics are expressed as means and standard deviations for continuous data. Data were tested for normality using the Shapiro–Wilk test. Data were analyzed based on sex and O3I categories. The O3I categories were defined as high risk, <4%; moderate risk, 4–8%; low risk, <8%. When data were distributed by both sex and O3I category, frequencies were reported. Differences in outcomes between groups were calculated using analysis of variance (ANOVA) or the Mann–Whitney U test. Relationships between diet and blood biomarkers were analyzed using Pearson’s correlations. Statistical significance was set a priori at *p* < 0.05.

## 3. Results

### 3.1. Participant Data

A total of 34 athletes (*n* = 19 male; *n* = 15 females) completed the dietary analysis and n-3 FA collection. Athlete demographics are presented in Table 1.

### 3.2. Erythrocyte Fatty Acids and Omega 3 Index (O3I)

Blood FA profiles (%) of male and female rugby players are shown in Table 2. There were no differences in EPA, DHA, ALA, or AA as a proportion of total erythrocyte fatty acids between groups. Surrogate markers of inflammation, n-6:n-3 FA and EPA:AA ratios were similar between groups (*p* > 0.05) (Figure 1).

The average O3I for all athletes was 4.54% ± 1.77 and ranged between 1.71 and 10.03%. Figure 2 details the individual O3I between male and female athletes. Male and female rugby players exhibited similar O3I values (*p* = 0.471), 4.48% and 4.61%, respectively. Athletes who reported taking an n-3 FA supplement had a higher O3I compared to those not taking a supplement (4.82% vs. 3.94%, *p* = 0.183). While the difference failed to reach statistical significance, those not taking a supplement were in the high-risk category (<4%) and those taking a supplement were in the intermediate-risk category (4–8%). When distributed by O3I category, there were a greater proportion of male and female rugby players in the high-risk category for those not taking a supplement (45.5% vs. 39.1%) (Table 3). However, 61% of rugby players taking a supplement were in the low- to intermediate-risk categories compared to 54.5% in those not supplementing.

### 3.3. Dietary Intake

Table 4 details the dietary intake of EPA, DHA, and ALA between male and female rugby players. There were no differences in EPA, DHA, and ALA consumption between males or females (*p* > 0.05). In the absence of supplementation, none of the athletes consumed the daily adequate intake levels of ALA (1.6 g·day^−1^ for men, 1.1 g·day^−1^ for women) [11] or the recommended levels of EPA + DHA (500 mg·day^−1^) [16]. With supplementation, male and female athletes consumed the adequate and recommended levels of ALA and EPA + DHA, respectively.

Dietary intake of EPA was positively correlated with the proportion of erythrocyte EPA (r = 0.434, *p* = 0.01) and O3I (r = 0.347, *p* = 0.045). DHA intake was positively correlated with erythrocyte EPA (r = 0.391, *p* = 0.02), weakly correlated with the proportion of erythrocyte DHA (r = 0.273, *p* = 0.12), and moderately correlated with O3I (r = 0.323, *p* = 0.06). Dietary ALA was not correlated with EPA, DHA, or O3I.

## 4. Discussion

We measured n-3 FA erythrocyte profile alongside analysis of n-3 FA dietary intake using a 21-item FFQ collected at the beginning of the 2017/2018 International Rugby 7s season in male and female elite Canadian players. To our knowledge, this is the first investigation examining n-3 FA status in an elite male and female rugby population. Main findings showed that no athletes had an O3I in the low-risk category of >8% without supplementation, and that dietary intake alone did not meet current n-3 FA dietary recommendations [11,17] including specific recommendations for elite athletes [17]. However, athletes taking an n-3 FA dietary supplement not only met but exceeded these population-based recommendations. Despite supplementation, O3I remained <8% for a majority of the cohort, plausibly suggesting that higher supplementation doses may be required to increase O3I scores in this specific athletic pool.

Similar data have been reported in athlete-based research [14,15,21,22]. In NCAA Division 1 athletes, dietary n-3 FA intake tends to be well below dietary recommendations [16], with EPA and DHA intakes averaging 100 mg·day^−1^ [14]. Other data reported that that only 9% of participants consumed 500 mg·day^−1^ of EPA and DHA, and only 4% met male and female ALA dietary recommendations [15]. North American diets continue to be low in EPA and DHA, with ALA being the principal n-3 FA consumed due to its presence in many commercially available vegetable oils [5,12,13]. ALA can be converted to EPA and DHA, and it has been shown that females are more efficient at this conversion then males, but with a higher percentage of this conversion going towards EPA vs. DHA [5,12]. Beyond consumption of n-3 FA, North American diets are high in n-6 fatty acids, specifically linoleic acid [29], which reduces ALA conversion rates and ultimately reduces stores of EPA and DHA [5,30].

O3I results have been examined in a few athletic populations, including some contact athletes [14,15,21,22]. A total of 34% of NCAA football players were considered high risk, 66% intermediate risk and no players were considered low risk with respect to O3I scores [22]. Von Schacky et al. reported O3I data in elite winter endurance athletes, with only one of 106 athletes having a low-risk O3I [21]. Out of our 34 athletes, only two (8.7%) were considered low risk, both being female. The majority, 52.9%, were intermediate risk and 41.2% were high risk. Mean O3I in our population was similar to previous athlete data, with males at 4.48 ± 1.12% and females at 4.61 ± 2.40%. Mean O3I in male football players was 4.4% ± 0.8% [22], 4.97 ± 1.19% in male and female winter athletes [21], and was 4.79% (4.37–5.25%) and 4.75% (4.50–5.01%) in male and female NCAA players, respectively [14]. It should be noted that our two females with low-risk O3I were also supplementing with n-3 FA. Despite a largely sub-optimal O3I, our cohort exhibited an n-6:n-3 ratio <5:1, which is not reflective of the typical Western-style diet. Although speculative, it appears that our cohorts diet quality may have been better than observed in previous studies conducted in Canadian adults [31]. Unfortunately, we did not capture dietary data on n-6 FAs. Interestingly, the athletes’ EPA/AA ratio was similar to that reported in Canadian adults [31]. While there is much debate surrounding the utility of fatty acid ratios and their association with health outcomes, the EPA/AA ratio seems to be a better, albeit understudied, marker of a more anti-inflammatory profile [32,33].

We also assessed n-3 FA supplementation in our population. Currently, there is no supplementation consensus for athletes, and further human research is warranted to examine the impact of n-3 FA supplementation in athletes, especially those participating in sports associated with high risks of sub-concussion head impacts and concussions. Within our cohort, 23 (68%) of the participants reported the use of an n-3 FA supplement, those supplementing showed higher dietary intakes compared to non-supplemented, and fewer participants were in the high-risk category for O3I. Ritz et al. also examined n-3 FA supplementation in NCAA athletes compared to O3I scores, with 15% of the population reporting the use of n-3 FA supplements and significantly higher O3I scores compared to athletes not taking a supplement [15].

Although previous research has indicated that concussion prevalence in rugby is lower than that of other contact-based sports [2], a more recent systematic review reported that concussion rates were the highest in rugby match play when compared to American football, soccer, and ice hockey [30]. Despite this, more research is needed to further elucidate this conundrum, particularly in women’s team sports. n-3 FA, particularly DHA, have shown promise in concussion recovery and neuroprotection [6,7,8,9], with much of the research performed in animal models demonstrating that a neuronal reduction in DHA post injury [34], plus a dietary induced brain DHA deficiency show significantly slower injury recovery, greater cognitive deficits, and anxiety-like behaviors [35]. Researchers have also examined the role of DHA in concussion as well as repetitive sub-concussive recovery, demonstrating that when provided prophylactically, it appears to attenuate the detrimental pathophysiological injury response [6,7,35,36,37,38,39]. There is limited human research studying the potential benefits of n-3 FA on head injury recovery [40,41,42]. A clinical trial was conducted on collegiate American football players to examine the potential neuroprotective effect of DHA supplementation, resulting in attenuated neurofilament light (NFL) levels throughout the course of the season [42]. With the extent of concussion risk in rugby, optimizing n-3 FA intake may be an important measure to support neuroprotection and concussion recovery.

Collection of retrospective dietary intake of n-3 FA along with supplement consumption through a FFQ allowed us to determine daily intake of EPA, DHA and ALA and compare this intake to O3I results. A major limitation of this project was that only one data point was observed to analyze erythrocyte n-3 FA and O3I results. Future research should examine n-3 FA intake, supplementation, and red blood cell (RBC) results strategically throughout the training and competitive season to look at changes in dietary and supplemental intake and their impact on RBC results.

## 5. Conclusions

We observed that elite male and female Canadian Rugby 7s players did not meet dietary recommendations for n-3 FAs and exhibited an unfavorable blood EPA and DHA profile. Further research is warranted to determine if this trend is seen throughout training and during a competitive season, and if controlled dietary supplementation improves blood EPA and DHA. Efforts should be made in this population to increase tissue EPA and DHA considering the importance of dietary n-3 FA, especially in athletes with an increased exposure to repetitive head impacts. Emphasis on promoting increased consumption of dietary sources of n-3 FA and/or the use of n-3 FA supplementation should be consideration. The risks of increasing EPA and DHA intake are quite low, with the potential benefits appearing to be substantial.

## Figures and Tables

**Figure 1 nutrients-13-03777-f001:**
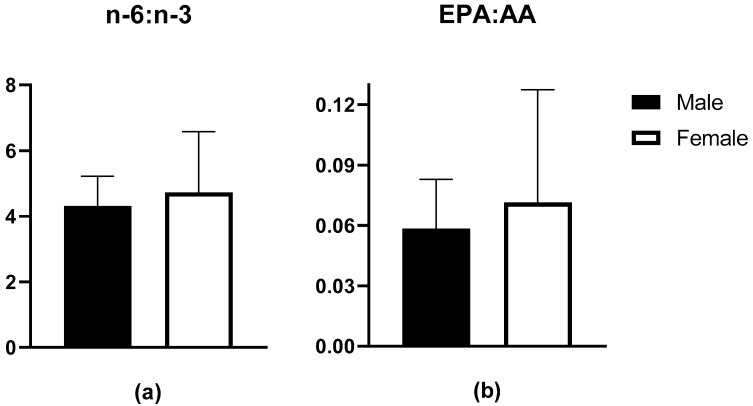
Erythrocyte fatty acid data for (**a**) n-6:n-3 ratio in males (*n* = 19, 4.31 ± 0.91) and females (*n* = 15, 4.73 ± 1.86); (**b**) EPA:AA ratio in males (0.06 ± 0.02) and females (0.07 ± 0.06).

**Figure 2 nutrients-13-03777-f002:**
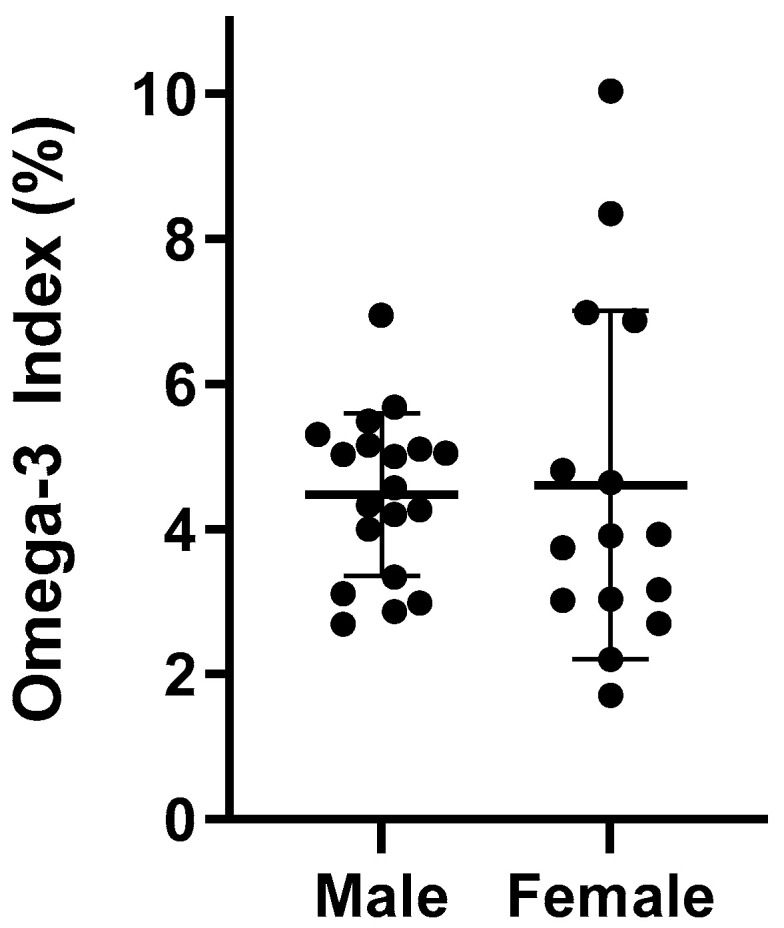
Individual O3I scores in males (*n* = 19) and females (*n* = 15).

**Table 1 nutrients-13-03777-t001:** Demographic characteristics of study participants based on sex (*n* = 34).

	Male (*n* = 19)	Female (*n* = 15)	*p*-Value
Age (y)	24.84 ± 2.32	23.45 ± 3.10	0.147
Height (cm)	185.80 ± 3.59	170.14 ± 6.47	<0.001
Weight (kg)	95.23 ± 6.93	71.21 ± 5.79	<0.001
BMI (kg/m^2^)	27.57 ± 1.67	24.61 ± 1.66	<0.001

**Table 2 nutrients-13-03777-t002:** Erythrocyte fatty acids (%) distributed by sex.

	Male (*n* = 19)	Female (*n* = 15)	*p*-Value
EPA	0.71 ± 0.28	0.87 ± 0.76	0.681
DHA	3.77 ± 0.94	3.74 ± 1.69	0.537
ALA	0.26 ± 0.11	0.27 ± 0.15	1.00
AA	12.41 ± 1.40	11.40 ± 2.79	0.319
O3I	4.48 ± 1.12	4.61 ± 2.40	0.471

**Table 3 nutrients-13-03777-t003:** O3I category based on reported supplementation status and sex.

	Male (*n* = 19)	Female (*n* = 15)	Total
With Supplementation			
Low risk	0 (0%)	2 (18.2%)	2 (8.7%)
Intermediate risk	9 (75%)	3 (27.3%)	12 (52.2%)
High risk	3 (25%)	6 (54.5%)	9 (39.1%)
Without Supplementation			
Low risk	0 (0%)	0 (0%)	0 (0%)
Intermediate risk	5 (71.4%)	1 (25%)	6 (54.5%)
High risk	2 (28.6%)	3 (75%)	5 (45.5%)

**Table 4 nutrients-13-03777-t004:** Dietary fatty acid intake with and without supplementation.

	Without Supplementation		With Supplementation	
	Male (*n* = 7)	Female (*n* = 4)	Male (*n* = 12)	Female (*n* = 11)
EPA (mg·day^−1^)	0.05 ± 0.03	0.07 ± 0.07	1272 ± 536	1189 ± 451
DHA (mg·day^−1^)	0.11 ± 0.07	0.16 ± 0.14	636 ± 268	595 ± 226
ALA (mg·day^−1^)	0.78 ± 0.78	0.11 ± 0.07	0.78 ± 0.78	0.11 ± 0.07

## Data Availability

Data can be made available on request.

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
