# Peer review of "An Evaluation of Omega-3 Status and Intake in Canadian Elite Rugby 7s Players"

_nutrients, 2021, doi:10.3390/nu13113777_

Round 1

Reviewer 1 Report

General comments: unclear why males and females were analyzed as separate groups when sex differences in O3I and O3 intakes are not a consideration in your purpose statement.  Provide rationale for why results were reported in this way. 

Given the link between O3I status and head trauma, was this asked of participants (recent history?)

FFQ evaluated consumption over past 6 months but was length of time athletes have been supplementing considered?

183-184: A stretch to say that females had higher O3I when p-value is .471. Suggest removal of this statement. 

188-193: What determines risk category? Explain how this was determined and/or provide citation. 

202-203: author cites no differences between groups. Do groups mean m/f or supplement/no supplement? If supplement, this seems unlikely given huge differences in all variables (EPA, DHA, ALA)

235-237: Discuss O6:O3 you found compared to recommendations, other studies. 

286-288: As written this does not match results found. It seems that elite athletes who do not supplement with O3 do not meet dietary recommendations. 

Author Response

Dear Reviewer,

Thank you for your careful consideration and evaluation of our manuscript. We have provided a detailed response below to your critiques and suggestions. We believe we have addressed all concerns related to the review. Thank you again, and we look forward to hearing from you.

Point 1: Unclear why males and females were analyzed as separate groups when sex differences in O3I and O3 intakes are not a consideration in your purpose statement.  Provide rationale for why results were reported in this way. 

Author’s reply: We’ve addressed this briefly:

“It is of particular interest to determine if there are sex differences associated with n-3 FA intake and status.” (Lines 86-87)

“A second objective of this study was to determine if there were any notable sex differences in n-3 FA intake and status.” (Lines 90-92)

Point 2: Given the link between O3I status and head trauma, was this asked of participants (recent history?)

Author’s reply: As noted in our introduction and subsequently in our discussion, head trauma is an ever-present concern for Rugby players' health and safety. Since accumulating evidence suggests that n-3 FAs play a role in the attenuation of head trauma, our goal for this study was to initially evaluate, for the first time, the typical dietary patterns related to n-3 FAs and blood n-3 FA status in elite Rugby players. As described in this manuscript, Rugby players exhibited poor intake of n-3 FAs and have sub-optimal n-3 FA levels. It is also to the author’s knowledge there is no study that has examined the risk ratio associated with head trauma and O3I status in humans or animal. Indeed, studies have demonstrated prophylactic treatment with n-3 FA attenuating the delirious effects of head trauma, a reduction in number of concussions or other head trauma incidents is not associated with a higher O3I. Though we collected medical information, including head trauma, we did not attempt to correlate those outcomes with current O3I or n-3 FA intakes given that we did not have historical dietary intake information beyond the 6 months period. We do plan to build upon the foundation established in this analysis by conducting a n-3 FA supplementation trial in this population, including head trauma data and brain biomarkers.

Point 3: FFQ evaluated consumption over past 6 months but was length of time athletes have been supplementing considered?

Author’s reply: We agree that gathering the most amount of information, especially dietary and supplementation practice that directly effects RBC n-3 status, would be ideal. However, it should be noted that the FFQ we used has been validated in comparison to blood fatty acid levels in two studies (Sublette, 2011 & Ritz, 2020), one of which was exclusively in an athlete population. Additionally, our dietary data was supplemented with RBC n-3 FA data used to corroborate and compare the findings.

Point 4: 183-184: A stretch to say that females had higher O3I when p-value is .471. Suggest removal of this statement.

Author’s reply: Although we acknowledged the difference was not statistically significant, we agree that we should re-word for consistency with the data.

“Male and female rugby players exhibited similar O3I values (p = .471), 4.48% and 4.61%, respectively.”  

Point 5: 188-193: What determines risk category? Explain how this was determined and/or provide citation. 

Author’s reply: The risk categories are discussed in the introduction (lines 75-78). However, to minimize confusion, we’ve opted to add this description in the methods as well.

“Data were analyzed based on sex and O3I categories. The O3I categories were defined as high-risk: <4%, moderate-risk: 4-8%, and low-risk: <8%.” (Lines 171-173)

Point 6: 202-203: author cites no differences between groups. Do groups mean m/f or supplement/no supplement? If supplement, this seems unlikely given huge differences in all variables (EPA, DHA, ALA)

Author’s reply: Thank you for pointing this out. We’ve updated for clarity as there was no difference based on sex.

“There were no differences in EPA, DHA, and ALA consumption between males or females (P > .05).” (Lines 212-213)

Point 7: 235-237: Discuss O6:O3 you found compared to recommendations, other studies. 

Author’s reply: We’ve included a short discussion on the ratios:

“Despite a largely sub-optimal O3I, our cohort exhibited an n-6:n-3 ratio <5:1 which is not reflective of the typical Western-style diet. Although speculative, it appears that our cohorts diet quality may have been better than observed in previous studies conducted Canadian adults [32]. Unfortunately, we did not capture data on n-6 FAs. Interestingly, the athletes’ EPA/AA ratio was similar to that reported in Canadian adults [32]. While there is much debate surrounding the utility of fatty acid ratios and their association with health outcomes, the EPA/AA ratio seems to be a better, albeit understudied, marker of a more pro-inflammatory profile [33,34].” (Lines 259-266)

Point 8: 286-288: As written this does not match results found. It seems that elite athletes who do not supplement with O3 do not meet dietary recommendations.

Author’s reply: Those taking a supplement did have a slightly higher O3I (4.82% v 3.94%); however, those taking n-3 FA supplements still appear to not have an “optimal” n-3 status. As such our observation still stands that, “Rugby-7s players did not meet dietary recommendations for n-3 FAs and exhibited an unfavorable blood EPA and DHA profile.”  

Reviewer 2 Report

Comments to work titled ”An Evaluation of the Omega 3 Status and Intake in Canadian Elite Rugby 7s Players”.

In the work, an issue significant for the health and exercise capacity of athletes is undertaken.

Minor comments:

1) For a more comprehensive characteristic of the group, the experience of the athletes under study could be given.

2) In the ‘Methods’ section, nutrition norms could be referenced, those to which the authors refer to in the evaluation of Omega 3 acid consumption.

3) I have not noticed statistical analysis of the data presented in Tab. 3.

4) The conclusions are too laconic and do not mirror all of the analysed areas and correlations – this requires revision.

5) I am in agreement with the authors’ opinion regarding the limitations of the work and further directions for research.

Author Response

Dear Reviewer,

Thank you for your careful consideration and evaluation of our manuscript. We have provided a detailed response below to your critiques and suggestions. We believe we have addressed all concerns related to the review. Thank you again, and we look forward to hearing from you.

Point 1: For a more comprehensive characteristic of the group, the experience of the athletes under study could be given.

Author’s reply: Thank you for this comment. Due to the specific population and location of the study, the Rugby players in the study may be readily identified. To protect and preserve the anonymity of the players we potent to describe their level of experience only briefly.

Point 2: In the ‘Methods’ section, nutrition norms could be referenced, those to which the authors refer to in the evaluation of Omega 3 acid consumption.

Author’s reply: Thank you for your comment. We have updated the methodology to address this.

“Average daily intake of n-3 FAs was used to determine if athletes met the IOC recommendation of 2 g·d-1 of n-3 FA from a food source or supplement or the much more defined Academy of Nutrition and Dietetics recommendations of 500 mg·day- DHA+EPA.” (Lines 124-127) 

Point 3: I have not noticed statistical analysis of the data presented in Tab. 3.

Author’s reply: Thank you for bringing this to our attention. We’ve updated a portion of the statistical analysis to include frequencies.

“When data were distributed by both sex and O3I category, frequencies were reported.” (Lines 173-174).

Point 4: The conclusions are too laconic and do not mirror all of the analyzed areas and correlations – this requires revision.

Author’s reply: Thank you for bringing this to our attention. We have made what we feel are appropriate revisions to the conclusion to further emphasis the key concern being low O3I scores and dietary intake in athletes participating in a contact sport where concussion risk is higher compared to other sport.

Point 5: I am in agreement with the authors’ opinion regarding the limitations of the work and further directions for research.

Author’s reply: Thank you for your comment